**Understanding solar geoengineering's potential to limit sea level rise requires attention from cryosphere**
**experts**
Peter J. Irvine[1], David W. Keith[1], John Moore[2,3]
1 - Harvard John A. Paulson School of Engineering and Applied Sciences, Cambridge, Massachusetts, USA
2 - Joint Center for Global Change Studies, College of Global Change and Earth System Science, Beijing Normal
University, Beijing 100875, China
3 - Arctic Centre, University of Lapland, Rovaniemi 96101, Finland
**Abstract**
Stratospheric aerosol geoengineering, a form of solar geoengineering, is a proposal to add a reflective layer of
aerosol to the stratosphere to reduce net radiative forcing and so to reduce the risks of climate change. Solar
geoengineering could reduce temperatures and so slow melt, but the efficacy of solar geoengineering at reducing
changes to the cryosphere is uncertain as is its ability reverse ice sheet collapse once initiated.  Here we review the
literature on solar geoengineering and the cryosphere and identify the key uncertainties that research could address.
Solar geoengineering may be more effective at reducing surface melt than a reduction in greenhouse forcing that
produces the same global-average temperature response. Studies of natural analogues and model supports this
conclusion. However, changes below the surfaces of the ocean and ice-sheets may strongly limit the potential of
solar geoengineering to reduce the retreat of marine glaciers. High-quality process model studies may illuminate
these questions. Solar geoengineering is a contentious emerging issue in climate policy and it is critical that the
potential, limits and risks of these proposals are made clear for policy makers.
**1. Future Sea-level rise and the potential of solar geoengineering**
How far sea-levels would rise under some scenario of future climate change depends mainly on global temperature
rise, and uncertainties in projections rise rapidly as warming increases more than 2°C above pre-industrial (Jevrejeva
et al., 2016; Kopp et al., 2014). Most of this uncertainty is due to a lack of agreement on how the large ice sheets
will respond (Bamber and Aspinall, 2013; Oppenheimer et al., 2016). For example, two recent high-profile
publications made conflicting estimates of Antarctica's contribution to sea-level rise by 2100 with a best-guess of
10cm (Ritz et al., 2015), and of around 1m (DeConto and Pollard, 2016).
A rapid transition towards a carbon-free economy will reduce additional temperature increases but the temperature
response to cumulative emissions—and thus the impact on sea level—will remain for millennia without measures
beyond emissions cuts (Clark et al., 2016). Two broad categories of measures might reduce long-term commitments
to global sea level rise: solar geoengineering and atmospheric carbon removal. Solar geoengineering which
describes a set of proposals to increase Earth's albedo, is not a substitute for emissions cuts. But it could offer an
independent means of temporarily reducing radiative forcing and thus the impacts of climate change, and so be a
complement to emissions cuts. The two responses may be synergistic: carbon removal can reduce the long-term
driver of climate change, while solar geoengineering might temporarily reduce the net radiative forcing. Our focus is
on assessing solar geoengineering impact on sea level rise because existing research is quite limited and because its
effects (per unit temperature change) may not be the same as those achieved by reducing temperature by de-
carbonizing.
The human, environmental and financial costs of sea level rise are substantial. The rapidly rising concentration of
population and infrastructure in coastal cities mean that costs of flooding without adaptation measures are projected
to be $50 trillion per year by 2100, while coastal protection would cost $15-70 billion per year (Hinkel et al., 2014).
One important consideration is that sea level rise is not globally uniform, due to a combination of local factors:
glacial isostatic adjustment and ground water extraction resulting in local vertical land movement; the self-
gravitational influence of mass loss from the large ice sheets; and changes in ocean dynamics and rates of volume
expansion of warming sea water. Taking all these together, Jevrejeva et al. (2016) find that the 80-90% of global
coastlines will experience sea level rises about twice as large as the global ocean average.
Whilst some, including one of us (Keith), have been working on solar geoengineering for decades, more than ten
times as many articles have been published on the topic since 2007 than before. Whilst many proposals for solar
geoengineering have been made, work now focuses on a few of the more likely candidates. Marine Cloud
Brightening, a proposal to increase the albedo of marine strato-cumulus by releasing sea-salt aerosols from ships
(Latham, 1990); Cirrus Cloud Thinning, a proposal to suppress cirrus cloud persistence, and hence reduce their
warming effect, by releasing ice nuclei to encourage the formation of larger, shorter-lived ice crystals (Mitchell and
Finnegan, 2009); and Stratospheric Aerosol Geoengineering, a proposal to release aerosol particles into the
stratosphere to create a persistent reflective aerosol layer scattering a small fraction of incoming light back to space
(Budyko, 1977). Of these proposals stratospheric aerosol geoengineering is the most likely to be technically
achievable. Multiple, independent feasibility assessments of the proposal conclude that a substantial cooling could
be achieved with a few Terragrams of material released per year and that lifting a Terragram to the lower
stratosphere (~20km) could be achieved at a cost of order one billion US dollars per Terragram (McClellan et al.,
2012; Moriyama et al., 2016). The clouds and aerosols chapter of the last IPCC report concluded that "there is
medium confidence that stratospheric aerosol [geoengineering] is scalable to counter the [radiative forcing] from
increasing [Greenhouse Gases (GHG)s] at least up to approximately 4 W m-2 [approximately the forcing a doubling
of $CO_2$ concentrations]" (Boucher et al., 2013). For this reason, here we focus on stratospheric aerosol injection and
unless otherwise stated, solar geoengineering will heretofore refer to stratospheric aerosol geoengineering only.
The tens of climate model studies of solar geoengineering prior to 2013 were summarized in the last IPCC report
(Boucher et al., 2013): "Models consistently suggest that [solar geoengineering] would generally reduce climate
differences compared to a world with elevated GHG concentrations and no [solar geoengineering]; however, there
would also be residual regional differences in climate (e.g., temperature and rainfall) when compared to a climate
without elevated GHGs." This reduction in the magnitude of many climate trends means that solar geoengineering
may offer a means to reduce the risks of climate change (Keith and Irvine, 2016).
Beyond its effect on climate (which will be discussed in more depth below), stratospheric aerosol injection would
have a number of side-effects (Irvine et al., 2016). Simulations of stratospheric sulphate aerosol injection (the most
commonly analyzed scenario of stratospheric aerosol geoengineering) consistently show that it would lower ozone
concentrations, delaying the recovery of the ozone hole by a number of decades (Pitari et al., 2014; Tilmes et al.,
2012). As well as scattering light back to space the stratospheric aerosol cloud would also scatter light downwards
shifting the balance of direct to diffuse light which could boost plant productivity though would reduce the
efficiency of concentrating solar power plants (Kravitz et al., 2012). The aerosols would also absorb radiation,
warming the stratosphere affecting stratospheric chemistry and dynamics (Tilmes et al., 2009). The magnitude of
these side-effects will depend on the properties of the injected aerosols, and alternatives to sulphate particles may
have substantially reduced side-effects (Keith et al., 2016).
In its seminal 2009 report (Shepherd et al., 2009), the United Kingdom's Royal Society predicted that the social and
political challenges posed by solar geoengineering would be far greater than the technical ones. Its potentially low
cost could mean that individual nations or very wealthy individuals could have the resources to deploy solar
geoengineering (Weitzman, 2014). The global impacts of any large-scale deployment could be the source of
international tension and poses a serious challenge for international governance (Victor, 2008).
Technical analyses and climate model simulations suggest solar geoengineering may offer a means of reducing the
risks of climate change but it would also introduce new risks, both physical and socio-political. A robust
understanding of the potential and limits of solar geoengineering as a means to reduce climate risks is a necessary,
but not sufficient, basis for a much broader discussion of this idea. This study aims to highlight the key questions
around the sea-level rise response to solar geoengineering that only the sea-level and cryosphere community will be
able to resolve. In section 2, we provide a brief review of studies into the sea-level rise response to solar
geoengineering noting the methodological shortcomings and gaps in the literature. In section 3, we evaluate how the
effects of solar geoengineering and a reduction in GHG forcing could on sea-level rise could differ, discussing its
potential effects on thermosteric sea-level rise, surface mass balance and on ocean-driven melt of ice-shelves and
discharge from marine glaciers. In the sub-section on surface mass balance we make an initial assessment on the
relative efficacy of solar geoengineering as seen in the Geoengineering Model Intercomparison Project (GeoMIP).
In section 4, we summarize the results briefly and make a number of recommendations for research.
**2. Critical review of existing literature on solar geoengineering and sea-level rise**
As solar geoengineering would reduce temperatures across the world, offsetting some of the warming from elevated
GHG concentrations, it is clear that to first order it would reduce both the thermal expansion of the oceans and the
melting of land ice. Wigley (2006), Moore et al. (2010) and Irvine et al. (2012) illustrate this using simple models of
the sea-level rise response to a range of solar geoengineering scenarios. Moore et al. (2010) used a semi-empirical
model relating radiative forcing to sea level calibrated by tide gauge data from the past 200 years to evaluate a range
of different forms of solar geoengineering. Wigley (2006) and Irvine et al. (2012) adapted the simple models used in
the Intergovernmental Panel on Climate Change third and fourth assessment reports, respectively, to evaluate a
range of different levels of cooling from solar geoengineering. Moore et al. (2015) used the relationship observed
between sea surface temperatures and Atlantic hurricanes to evaluate the effects of solar geoengineering on storm
surges along the East coast of North America.
In addition to these studies with models of reduced complexity there have been a few studies employing glacier and
ice sheet models. Irvine et al. (2009) conducted a study of the response of the Greenland Ice Sheet to a range of
idealized and fixed scenarios of solar geoengineering deployment using the GLIMMER ice dynamics model driven
by temperature and precipitation anomalies from a climate model and found that under an idealized scenario of
quadrupled $CO_2$ concentrations solar geoengineering could slow and even prevent the collapse of the ice sheet.
Applegate and Keller (2015) used a simplified ice dynamics model driven by an Earth system model of intermediate
complexity to evaluate the response of the Greenland Ice Sheet to scenarios of future GHG emissions and solar
geoengineering deployment. They found that whilst solar geoengineering could slow or halt melting, there is strong
hysteresis and restoring temperatures would not lead to a rapid recovery of the ice sheet. Zhao et al. (2017) evaluate
the response of the 94,000 High Mountain Asia glaciers using an empirical model based on each glacier's median
elevation sensitivity to changes in only temperature and precipitation. Under scenarios where solar geoengineering
halts regional temperature increases, 30% of present-day glaciated area will still be lost this century due to the
glaciers being out of balance with present day climate.
These studies illustrate that if solar geoengineering were deployed it could reduce the rate of sea-level rise
substantially compared with greenhouse forcing alone. However, all studies to date have employed simplified global
models. Thus these studies miss out on some of the fundamental differences between scenarios of climate change
with and without solar geoengineering.
Whilst increasing the planetary albedo would undoubtedly cool the climate, the effects of a reduction in incoming
light differ substantially from the heat-trapping effects of greenhouse gas forcing. GHG forcing acts more-or-less
uniformly, whereas solar forcing acts only when the sun is up. Offsetting the GHG forcing with solar forcing would
therefore produce seasonal, diurnal and latitudinal differences in radiative forcing.
Furthermore, solar forcing acts primarily on the surface whereas GHG forcing acts most strongly on the middle
troposphere where infrared radiation escapes to space. As a result, solar forcing reduces the intensity of the
hydrological cycle more strongly than does a reduction in GHG forcing that produces the same top-of-the-
atmosphere radiative forcing. Bala et al. (2008) evaluated the sensitivity of the global hydrological cycle, finding a
2.4 %K$^{-1}$ change in global mean precipitation for solar forcing and only a 1.5 %K$^{-1}$ for $CO_2$ forcing. They note that
insolation changes result in relatively larger changes in net radiative fluxes at the surface than $CO_2$ forcing resulting
in larger changes in sensible and latent heat fluxes.
Beyond this fundamental difference in the climate response to solar forcing, some stratospheric aerosols, particularly
sulfuric acid the most important single proposal, have significant near infrared absorption bands that would result in
a warming of the stratosphere. This warming would have dynamic implications, for example McCusker et al. (2015)
find significant changes in circulation in the Antarctic stratosphere which propagates down to affect surface winds
and the mixing of waters around Antarctica..
These differences between greenhouse gas and shortwave forcing matter for making predictions of the surface mass
balance of glaciers and ice-sheets: Melting of ice peaks during the day in summer when it is most sensitive to
changes in surface energy balance; Changes in snowfall amount and seasonality would affect glacier mass balance;
And, solar geoengineering would alter atmospheric and oceanic circulation patterns which can affect the upwelling
of warm waters around ice shelves, weakening them. In the following sections we will identify how solar
geoengineering could affect these factors and identify the most pressing uncertainties.
### 3.   Response of sea-level rise to solar geoengineering
In this section we evaluate the potential effects of solar geoengineering on the various contributions to sea-level rise,
addressing thermosteric sea-level rise, surface mass balance, and ice-shelf collapse and dynamic mass loss. In
making this evaluation we aim to bring light to two overarching questions:
• How effective is solar geoengineering at reducing a given contribution to sea-level rise as compared to a
reduction in GHG forcing that produced the same global-average change in temperature? Would, for
example, one Celsius of global average cooling from solar geoengineering lower the surface-mass-balance
contribution to sea level rise by more or less than would one Celsius of cooling achieved by reduced GHG
forcing?
• What fundamental limits are there to the potential for solar geoengineering to reduce or reverse sea-level
rise? That is, in what ways do the contributions to sea-level rise exhibit hysteresis or tipping points that
would make halting or reversing sea-level rise with solar geoengineering more difficult than may be
expected?
### 3.1. Thermosteric Sea-level rise
Global thermosteric sea-level rise is the simplest contribution to global sea-level rise. Thermosteric sea level can be
computed from the density profile over depth, which is derived from temperature and salinity data, (Dangendorf et
al., 2014). Changes in temperature dominate steric sea level variability. A reduction in total radiative forcing no
matter if it comes from a reduction in GHG forcing or from solar geoengineering, will produce the same reduction in
heat transfer to the ocean and so the same reduction in thermosteric sea-level rise.
Bouttes et al. (2012) explore the reversibility of thermosteric sea-level rise using a coupled climate model for a
range of $CO_2$ ramp-up and ramp-down scenarios, though the results apply equally to the case of solar
geoengineering. They find that the thermosteric sea-level rise response to their scenarios can be roughly
approximated by the integral of radiative forcing which closely corresponds to the total heat uptake of the oceans
over the simulations. This implies that to halt thermosteric sea-level rise, radiative forcing would need to be restored
to pre-industrial conditions. As the total forcing is ramped down, the warmed oceans become out of equilibrium with
the now-cooled atmosphere and slowly give off the heat they absorbed, gradually reversing the thermosteric sea-
level rise that had occurred during the ramp-up (See figure 1 of Bouttes et al. (2012)).

**3.2. Surface Mass Balance**

Many ice-sheet and glacier models use a simple parameterization of surface mass balance, using a positive degree-
day factor to estimate the amount of melt per degree above freezing at the glacier surface (Ohmura, 2001). Degree
day factors are determined empirically and vary due to surface albedo, meaning that a weathered ice surface such as
the Greenland ice margin are rather dark and have high degree-day factors, while pristine snow cover has a low
factor. This degree-day approach has been used in all studies of solar geoengineering's effect on surface mass
balance to date but it has some important limitations.
Fundamentally the surface melt rate depends on the availability of energy at the surface; this means that net
shortwave, net longwave, sensible and latent fluxes all matter. Despite only accounting for temperature, degree-day
approaches generally produce similar results to more complete energy balance models for surface melt, this is
because downwelling longwave, which typically is the dominant contributor to the energy flux, correlates well with
surface air temperature since much of the downwelling longwave is emitted in the first 1 km of the atmosphere
(Ohmura, 2001). However, degree-day approaches cannot capture the full response to changes in energy fluxes and
a look at some case studies reveals that changes in insolation can have outsized impacts which will be under-
estimated by degree-day approaches.
Increased summer insolation at high-latitudes during the Eemian interglacial period (115-130 kyr BP) raised
temperatures but also directly affected surface melt. Van de Berg et al. (2011) made an attempt to separate the
contributions of elevated temperatures and increased solar forcing and suggested that 45% of the change in surface
mass balance could be attributed to the changed solar forcing alone.
Volcanic eruptions provide a more contemporary analogy to the potential effects of solar geoengineering on surface
melt. Fettweis et al. (2007) simulated the surface mass balance of Greenland between 1979 and 2006 and find
maxima for surface mass balance in 1983 and 1992, the years after the El Chichon and Pinatubo eruptions,
respectively. Hanna et al. (2008) combine observations and modeling to evaluate the surface mass balance of
Greenland over a longer period finding that the years following El Chichon and Pinatubo have the third lowest and
the lowest runoff, and the third and sixth greatest surface mass balance, respectively between 1958 and 2006.
In an analysis of recent changes over Greenland, Hofer et al. (2017) found that the substantial reduction in cloud
cover over Greenland in the past two decades is the likeliest cause for the accelerated mass loss from the ice-sheet
over this period. To arrive at this result they simply calculated how much melt would result from the change in
downward surface shortwave energy received over the melt season as a result of the change in cloud cover, and
compared this against the other contributions to melt and accumulation. They find that the ~10% reduction in
summer cloud cover over Greenland in the past two decades led to a ~4000 Gt loss of mass making it the dominant
driver of surface mass balance change in this period. In Svalbard the opposite has been seen, with less melt than
projected by degree-day models of glacier mass balance due to an increase in cloud cover partially offsetting the
increased temperatures Slangen et al. (2016). Giesen and Oerlemans (2013) and Lang et al. (2015) use glacier mass
balance models that account for this change in surface shortwave and produce a better fit to observations.
These examples suggest that solar geoengineering could be more effective at changing surface melt than achieving
the same reduction in temperature with a reduction in GHG forcing. To evaluate the differences in the drivers of
surface mass balance we conduct a simple analysis of the well-studied GeoMIP G1 experiment, in which the
radiative forcing from an instantaneous quadrupling of $CO_2$ concentrations is offset by a reduction in the solar
constant sufficient to restore the pre-industrial radiative balance and global-mean temperature (Kravitz et al. 2011).
Kravitz et al. (2013) provide an overview of the climate response to this experiment from 12 Earth System Models,
and we analyze data for these same 12 models.
The models that ran the GeoMIP G1 experiment did not perfectly restore global-mean-temperatures to the pre-
industrial, although the differences in top of atmosphere radiative forcing were specified to be less than $0.1 Wm^{-2}$. As
we are interested in the relative efficacy of solar geoengineering compared to an equivalent reduction in $CO_2$ forcing
it is necessary to rescale these results so that they match the models' pre-industrial global-mean temperature.
$$F = \frac{\left(GMT_{4xCO_2} - GMT_{control}\right)}{\left(GMT_{4xCO_2} - GMT_{G1}\right)}$$

Where, F is the ratio between the global-mean temperature (GMT) anomaly of $4xCO_2$ - control and of $4xCO_2$ – G1.
This ratio is greater than 1 if G1 is warmer than the control and less than 1 if it cooler than the control. This ratio can
then be used to rescale the effects of the reduction in solar constant to produce a synthetic scenario G1* in which
global-mean temperatures would be identical to the control case:
$$X_{G1^*} = X_{4xCO_2} + F \times \left(X_{G1} - X_{4xCO_2}\right)$$

Where X is the variable to be rescaled. We apply this equation to all variables in our analysis. We also generate
scenarios where regional, annual-mean temperatures are restored using the same approach (G1-Greenland and G1-
Antarctica).
Figures 1 and 2 compare the regional-mean anomalies from the control for the $4xCO_2$, G1* and G1-local
experiments, and the "efficacy" of G1* and G1-local at offsetting $4xCO_2$ trends for Greenland and Antarctica,
respectively. Efficacy is defined as the fraction of the $4xCO_2$ trend offset:
$$E = \frac{X_{4xCO_2} - X_{Geo}}{X_{4xCO_2} - X_{control}} \times 100\%$$

As an example, many studies have shown that solar geoengineering is more effective at offsetting global-mean
precipitation than global-mean temperature. Tilmes et al. (2013) find that compared to the control the GeoMIP
ensemble mean showed a 6.9% increase in global-mean precipitation in $4xCO_2$ and a 4.5% reduction in G1, taking
these numbers we find an efficacy of 165%, that is whilst 100% of the global-mean temperature response has been
offset, 165% of the global precipitation response has been offset. When comparing the global-mean temperature and
local-mean temperature efficacies we find if 100% of the global-mean temperature has been offset, 90% of the
Greenland mean temperature has been offset (90% efficacy relative to global temperature) and if 100% of the
Greenland-mean temperature has been offset 111% of the global-mean temperature has been offset (111% efficacy
relative to local temperature).

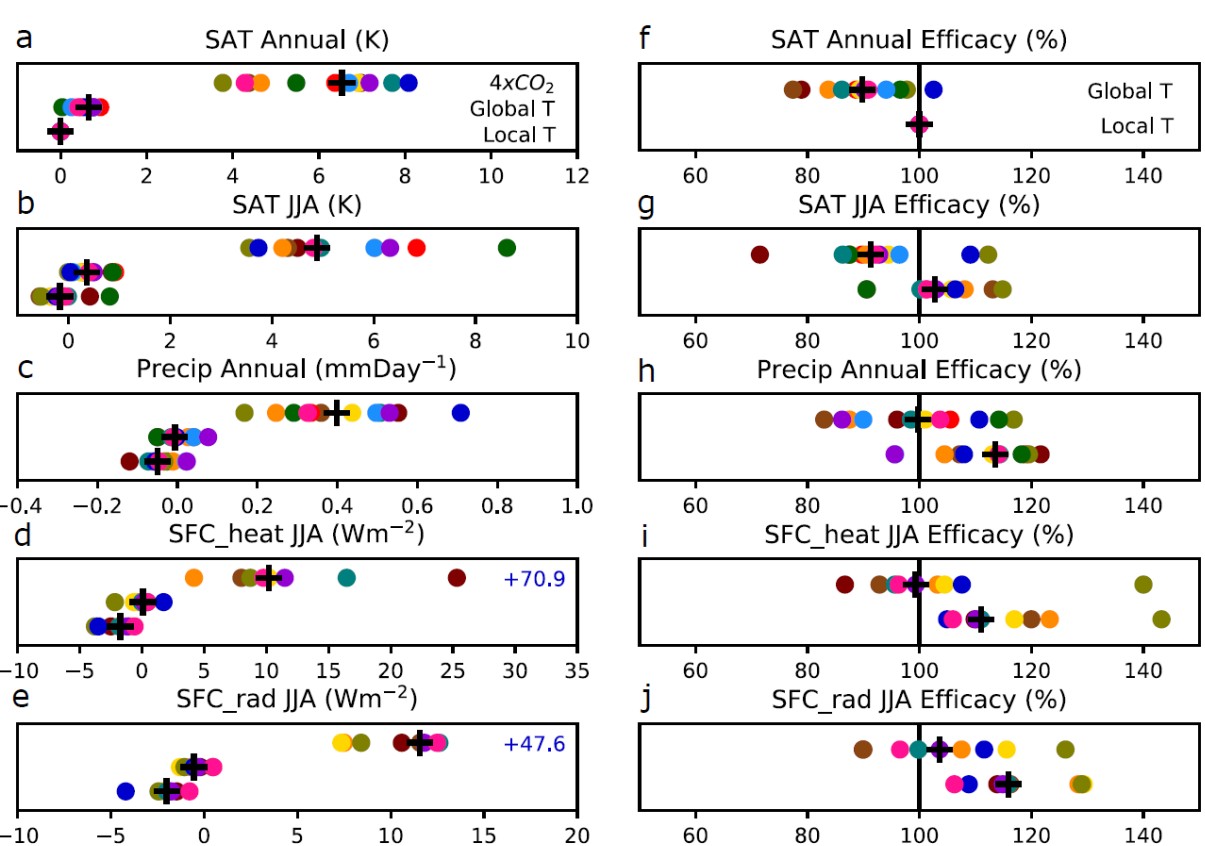

Figure 1. Regional-mean anomalies (left) and efficacies (right) of G1* and G1-Greenland at offsetting $4xCO_2$ – Control regional-mean anomalies for Greenland for each model within the GeoMIP G1-ensemble. On the left panel, the upper points show the $4xCO_2$ – Control anomaly, the middle row of points show the G1* results which restore global mean temperature, and the lower points show the results for G1-Greenland which restores local temperature. The ensemble median is shown with a plus symbol. The results from some outlier points have been displayed as text in the colour of the corresponding model. SFC_heat is the net heat flux into the surface, i.e. net SW + net LW – sensible heat – latent heat, and SFC_rad is the net radiative flux into the surface, i.e. net SW + net LW. Efficacy is defined in the text. Where data was unavailable these models have not been plotted for those variables.

In Greenland (Figure 1), G1* offsets most of the effects of $4xCO_2$, bringing climate much closer to the control conditions with a median efficacy that is within 10% of 100%. However, this result is a combination of G1* being under-effective at offsetting local temperatures, offsetting 90% of the annual-mean and 91% of the summer-mean, and being over-effective at offsetting the other fields relative to local temperatures, as seen in G1-Greenland results. There is a wide range of annual-mean precipitation responses across the ensemble in G1* but the ensemble median is close to 100%, i.e. the substantial increase in precipitation in $4xCO_2$ has been offset. The global-mean hydrological cycle has been weakened substantially but it seems local temperatures have been the dominant driver of the local hydrological response. The ensemble median shows a large increase in net downward surface radiation and surface heat flux, of greater than 10 Wm$^{-2}$ for the $4xCO_2$ – control anomaly, though some models show considerably larger changes. Relative to local temperature change, solar geoengineering is over-effective at

offsetting these changes in all models, with the ensemble median offsetting 116% of the net downward surface
radiation and 111% of the net downward surface heat flux increases that were seen in $4xCO_2$. These results suggest
that positive degree day melt schemes which do not account for these radiation and energy flux changes could
under-estimate the effectiveness of solar geoengineering at offsetting melt in Greenland by approximately 10%.

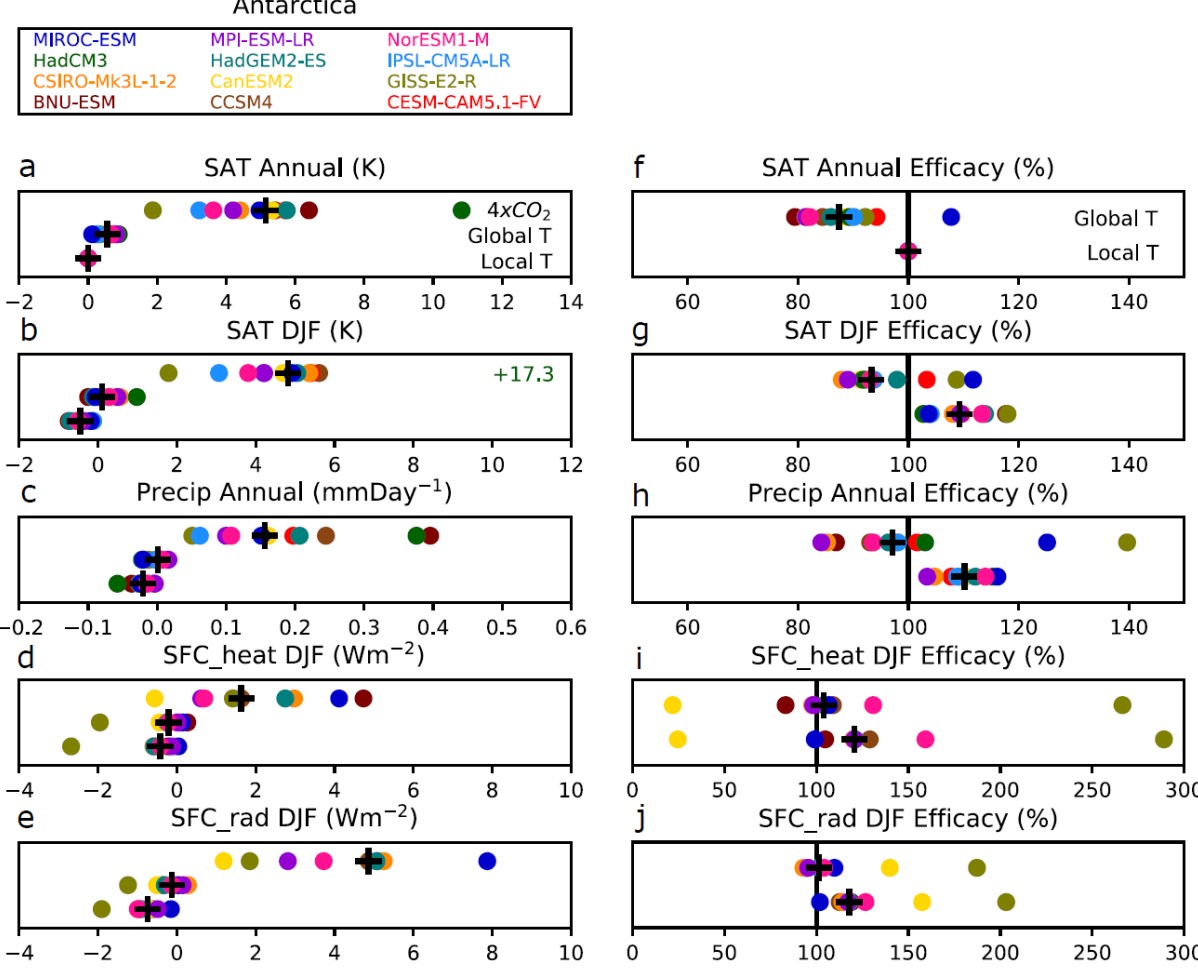

Figure 2. As Figure 1 but for Antarctica and Antarctic summer.

In Antarctica (Figure 2), A similar picture emerges as for Greenland with G1* being under-effective at offsetting
local temperatures, but, relative to local temperature change being over-effective at offsetting the other fields.
However, the implications of these results are different as melt plays only a small role in Antarctic surface mass
balance, with accumulation dominating and with the surface mass balance contribution of Antarctica to future sea-
level rise projected to remain negative for the foreseeable future. Ligtenberg et al. (2013) predict an increase of
Antarctic surface mass balance of 98 Gt year$^{-1}$ K$^{-1}$ using the RACMO2 model and Lenaerts et al. (2016) predict an
increase of 70 Gt year$^{-1}$ K$^{-1}$ using the CESM model. The ensemble median precipitation response is close to control
values in the G1* experiment, though there is substantial model spread, which suggests that regional temperatures
dominate the Antarctic hydrological response rather than the state of the global hydrological cycle which is
significantly weaker in G1*. These results suggest that the negative contribution to sea-level rise of the positive
surface mass balance response of Antarctica to global warming would decline roughly in line with temperatures if
solar geoengineering were deployed though more work is needed to explore this issue.

This simple assessment supports the view that solar geoengineering would have a greater potential to reduce surface
melt, and hence the sea-level rise contribution from surface mass balance changes of glaciers and the ice-sheets, than
previous studies have suggested. However, several factors would need to be accounted for in future work to make a
robust estimate of the efficacy of solar geoengineering at offsetting surface melt. Firstly, the impacts of a reduction
in incoming sunlight will be greater where the albedo of ice is lowest. A large and growing fraction of the ablation
zone of Greenland in summer is darkened by distributed surface impurities and snow algae revealed when the snow
layer is melted, these darkened areas typically have an albedo half that of clean ice (Ryan et al., 2018). The impact
of reduced sunlight will also be greater in low-latitude regions where the shortwave flux makes up a greater fraction
of the total contribution to the surface energy flux, e.g. in High Mountain Asia. For tropical and mid-latitude
glaciers, changes in accumulation due to changes in precipitation will also be an important factor to consider.
The results described here apply to a uniform reduction in incoming sunlight but the response to other, more realistic
forms of solar geoengineering could be tailored to produce different outcomes. For example, whilst a uniform
reduction in incoming sunlight would not offset all warming at high latitudes, stratospheric aerosol geoengineering
could be deployed to produce a thicker aerosol cloud at high latitudes to reduce high latitude temperatures in line
with global mean temperatures or to cool them further (Dai Z. et al., 2018; Kravitz Ben et al., 2018). However, it is
important to note that the effects of solar geoengineering cannot be limited to the area of application and there would
be remote impacts even if stratospheric aerosol geoengineering was limited just to polar regions (Robock et al.,
296 2008)

**3.3. Ice-shelf collapse and dynamic mass loss**

The other mechanism by which ice-sheets lose mass is by calving icebergs from marine-terminating glaciers and
here the effects of solar geoengineering are harder to anticipate. The rate of rate of discharge depends on how fast
the ice flows across the grounding line. The rate of ice flow depends on several factors that are affected by changes
in climate. Warmer ice is less viscous, allowing it to flow faster, though this is changing only very slowly and is
negligible for the ice sheets on centennial time scales (Slangen et al. 2016). Increased melt-water can penetrate to
the bed of the glacier and lubricate it, which may speed up the flow, although this "Zwally effect" seems not
especially important in Greenland where surface melt waters are efficiently drained in channelized drainage systems
such that changes in surface runoff have little impact on basal friction (Fleurian et al., 2016), and in Antarctica
surface melt is not as yet significant in fast-flowing glaciers (Joughin et al., 2009). For Antarctica where ice
discharge is the dominant loss mechanism, the most significant effect of climate change is to thin and weaken ice-
shelves which provide a buttressing effect, pushing back against the glaciers slowing their flow into the ocean.
Antarctica is so cold that little surface melt occurs on the ice-shelves, however relatively warm waters have been
observed penetrating below the ice shelves, melting them from below (Pritchard et al., 2012). The water mass
responsible for this melt is not the surface water around Antarctica, but rather the circumpolar deep waters
(originating around 500 m below the surface) that surround Antarctica. Surface winds have acted to pump this
relatively warm circumpolar deep water up and into the ice-shelf cavities. Here this relatively warm water can reach
the grounding line where the ice starts to float and where pressure requires the ice to have the lowest melting point
temperature. This ocean-driven melt has been observed to be thinning ice shelves, at rates as large as 50 m per year
at the grounding line and as high as 14 m per year averaged over the some of the larger ice shelfs (Rintoul et al.,
2016), weakening their buttressing effect and increasing the rate of discharge of glaciers into the ocean (Favier et al.,
2014). It is generally believed that the fate of the ice-shelves is likely to be determined by the degree to which this
circumpolar deep water is able to penetrate into the deep ice shelf cavities rather than by surface melt (Liu et al.,
2015; Pritchard et al., 2012).
A recent study (DeConto and Pollard, 2016), has challenged this view suggesting that the atmospheric warming that
led to the break-up of some Antarctic Peninsula ice shelves would, if the warming continued, destabilize the larger
southern ice shelves in the future. The process is through the hydrostatic head of melt-water filled crevasses which
results in "hydrofracture" and the rapid disintegration of the ice shelf (Scambos et al., 2013). Furthermore, they
suggest that once large ice shelves begin to retreat, the large unstable ice cliffs formed could promote further rapid
retreat, in a process dubbed marine ice-cliff instability (Pollard et al., 2015). Together these processes combined to
produce a substantially greater Antarctic contribution to sea-level rise than seen in earlier studies which did not
account for these highly uncertain processes (DeConto and Pollard, 2016).
Climate change and solar geoengineering will affect the ice-shelves, and hence the rate of discharge of marine
glaciers, primarily by changing surface air temperature and wind patterns that affect the upwelling of circumpolar
deep water. Solar geoengineering could lower surface air temperatures and hence reduce the likelihood of surface-
melt-induced hydrofracturing of the ice-shelves as assessed by DeConto and Pollard (2016). Whilst solar
geoengineering could lower surface air temperatures and surface ocean temperatures around Antarctica this would
have limited impact on the temperature of the deep circumpolar water mass responsible for thinning the ice-shelves
in the near-term as it is deep below the surface. As noted above, ocean-driven melt is primarily controlled by the
upwelling of these deep waters which is driven by Southern Ocean winds. A recent study of the effects of
stratospheric sulphate aerosol geoengineering in a scenario of future GHG emissions found that it would warm the
stratosphere, changing both atmospheric and oceanic circulation patterns (McCusker et al., 2015). They simulated a
greater upwelling of circumpolar deep-water relative to a scenario without an increase in GHG forcing, but that
ocean temperatures were significantly lower than in the GHG only scenario. If this result proves robust then it
suggests that whilst stratospheric aerosol geoengineering—or at least geoengineering using aerosols like sulfates
which strongly alter stratospheric heating rates—could lower surface melt considerably it may have a limited ability
to reduce ice shelf basal melt rates.
The dynamical response of marine glacier ice flow to changes in the buttressing effect of ice shelves is not simple
and there is the potential for runaway responses which would limit solar geoengineering's potential to slow or
reverse this contribution to sea-level rise. Fürst et al. (2016) show that ice shelves in the West Antarctic Amundsen
and Bellingshausen seas are extremely sensitive to calving, meaning that even small amount of increased calving
will trigger dynamical responses in the feeding ice streams increasing their flow rate. Furthermore, West
Antarctica's geography makes its ice sheet especially vulnerable to such changes. Much of the ice sheet rests on
bed-rock below sea-level which gets deeper further from the coast. This arrangement makes many of Antarctica's
glaciers susceptible to "marine-ice sheet instability" (Mercer, 1978), in that if the boundary layer begins to retreat,
the ice flow across the grounding line increases, prompting a self-sustaining retreat that would continue until a
bedrock ridge further inland. In fact, observations suggest that recent increases in the temperature of water around
Antarctica may have already triggered a process that will lead to the collapse of the Pine island and Thwaites
glaciers (Favier et al., 2014; Joughin et al., 2014). Unless an ice stream has exceptionally strong lateral buttressing
(Robel et al., 2016), a marine ice sheet instability, once started, may only be stopped by modifying bathymetry to
provide extra buttressing, as simulated by flow-band modeling on Thwaites glacier (Wolovick and Moore, 2018).
However, initial results from the BISICLES model evaluating the response of an idealized vulnerable marine glacier
to imposed warming found that returning to cooler conditions reversed the retreat that had begun during the
warming (Asay-Davis et al., 2016). It seems reasonable to expect that solar geoengineering may help to prevent
other marine glaciers from becoming unstable by limiting surface melt that could lead to ice-shelf collapse (as
emissions cuts would). It may be that significant losses from some West Antarctic glaciers are unavoidable by
simply returning climate and oceanic driving conditions to the pre-industrial and perhaps that even doing so would
not be sufficient to arrest the retreat.

## 4. Recommendations for research

In this study we've reviewed the literature on the effects of solar geoengineering on sea-level rise and highlighted
several gaps and shortcomings in the approaches used to date. We've also highlighted important differences between
a reduction in GHG forcing and solar geoengineering that will affect the surface mass balance of glaciers and ocean-
driven melt of ice-shelves and so the discharge rate of marine glaciers. We conclude with specific research
recommendations that will help to address the key questions we've highlighted earlier: Would solar geoengineering
be more, or less, effective at offsetting sea-level rise than an equivalent reduction in GHG forcing? And what are the
limits to solar geoengineering's potential to reduce or reverse sea-level rise?
4.1. Evaluate the sea-level rise response to scenarios of solar geoengineering deployment alongside other scenarios
of future climate change
Many of the new Earth System Models taking part in CMIP6 include coupled ice-sheet model components and are
ideal for making an initial assessment of the questions we have raised. The Ice-Sheet Model Intercomparison Project
phase 6 (ISMIP6) aims to evaluate the ice-sheet response of coupled ice-sheet models to idealized and future
emissions scenarios (Goelzer et al., 2018). The future emission scenario chosen by this project is the business-as-
usual SSP5-8.5 scenario (which reaches 8.5 $Wm^{-2}$ by 2100) which is also the basis for the GeoMIP6 G6 experiment
where the radiative forcing is reduced to match the SSP4-6.0 scenario (6.0 $Wm^{-2}$ by 2100) out to 2100. We
recommend that groups participating in both ISMIP6 and GeoMIP6 take this opportunity to extend the ISMIP6
protocol to the GeoMIP G6 experiment, i.e. producing a run including the coupled ice-sheet model and running an
offline ice-sheet model, to explore the effects of solar geoengineering on sea-level. To evaluate the relative efficacy
of solar geoengineering these results could be compared to the coupled ice-sheet model response to the SSP4-6.0
scenario which has a reduction in GHG forcing equivalent to that offset by stratospheric aerosol geoengineering in
GeoMIP6 G6.
Insight into the limits of solar geoengineering as a means of reducing sea-level rise can also be gained by extending
the idealized simulations studied in ISMIP6. ISMIP6 also focuses on an idealized simulation in which $CO_2$
concentrations rise at 1% per year until $4xCO_2$ is reached (after 140 years), we recommend extending this protocol
by fixing $CO_2$ concentrations at $4xCO_2$ values thereafter but also lowering the solar constant at such a rate that
global-mean temperatures are restored to control conditions after 140 years. We note that the Carbon Dioxide
Removal MIP also includes a similar experiment which reduces $CO_2$ concentrations at the same rate that they were
raised and would be an interesting target for study (Keller et al., 2018). These idealized ramp-up, ramp-down
scenarios would provide a solid basis for evaluating the potential of solar geoengineering, and carbon dioxide
removal, to reverse sea-level rise, showing the extent to which hysteresis and threshold behaviors would limit this
potential. Furthermore, a comparison between the solar constant and $CO_2$ ramp-down scenarios would allow an
evaluation of whether solar geoengineering would be more or less effective at reversing sea-level rise.
4.2. Evaluate the surface mass balance response to solar geoengineering using dedicated regional surface mass
balance models
As we show above, there are good theoretical reasons and now some limited model evidence to support the view
that solar geoengineering would be more effective than an equivalent reduction in GHG forcing. However, there are
several unknowns that preclude making any quantitative statements about this effect. For example, the steep
orography of the ablation zone will not be well-captured in coarse models, changes in surface albedo due to
impurities may not be well captured, and regional biases in climate can have a significant impact on results. We
therefore recommend that the analysis of the coupled ice-sheet models recommended above be complemented by
simulations with dedicated regional surface mass balance models. As noted above, a comparison between the
surface mass balance in the GeoMIP G6 and SSP4-6.0 scenarios would allow a quantification of the relative efficacy
of solar geoengineering at offsetting the reduction in surface mass balance in a warmer world.
4.3. Evaluate the effect of solar geoengineering on the upwelling of Antarctic Circumpolar Deep Water and on the
stability of the ice-shelves and marine glaciers.
The study of McCusker et al. (2015) suggests that stratospheric aerosol geoengineering may promote upwelling as
changes in stratospheric circulation could propagate downwards to change surface winds around Antarctica. If this is
the case, stratospheric aerosol geoengineering could be significantly less effective than a reduction in GHG forcing
at offsetting the increased upwelling of circumpolar deep water around Antarctica. Future work should investigate
whether this result is robust across the ensemble of models running the GeoMIP6 G6 stratospheric aerosol
experiment. In addition, as the climate response to stratospheric aerosols depends strongly on the type of aerosol
released and the distribution of the aerosols (Dykema et al., 2016), whether it may be possible to avoid unfavorable
wind patterns by deploying stratospheric aerosol geoengineering differently should be explored in further climate
model simulations.
4.4. Evaluate sea-level rise risks as part of an interdisciplinary evaluation of solar geoengineering
Sea-level rise is one of the key risks of climate change and so it will be important to understand the potential
efficacy and the limits of solar geoengineering as a means of reducing sea-level rise, however sea level rise is only
one of many issues that must be considered when discussing solar geoengineering. There are likely good reasons not
to deploy solar geoengineering with the objective of halting or reversing sea-level rise as this seems likely to require
a substantial reduction in global temperatures which could result in potentially harmful shifts in regional climate and
significant non-climatic side-effects (Irvine et al., 2012). Furthermore, whilst an understanding of the potential
physical consequences of climate change and solar geoengineering is necessary for a discussion of the potential use
of solar geoengineering, it is not sufficient. Whether and how to deploy solar geoengineering is a question that
demands a nuanced discussion encompassing not only the physical consequences of deployment but also a careful
consideration and negotiation of the complex socio-political issues it raises. A good understanding of the potential
and limits of solar geoengineering to reduce sea-level rise will be an important part of the foundation of this much
broader discussion which we hope the cryosphere research community will engage.

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
