# Peer review of "Understanding solar geoengineering's potential to limit sea level rise requires attention from cryosphere experts"

_The Cryosphere, 2017_

## Referee Comment (RC1) · Anonymous Referee #1 · 5 Feb 2018

This article reviews the links between solar engineering and the surface mass balance of glaciers and ice sheets. Given the potential importance of the topic I am rather reluctant to report this article to be a rather awkward read. It pokes out in many directions, but not sufficiently far enough in any one to be truly novel. Perhaps this opinion reflects that I am well-read on the general topic, and personally feel that this qualitative discussion on cryospheric implications fall short of The Cryosphere community's consistent ability to deliver quantitative assessments on just about every other front. Personal opinion aside, this article objectively resurveys many of the same well-trodden roads of Irvine et al. [2017; Earth Future], Keith and Irvine [2016; Earth Future] and Irvine et al. [2012; Nature Climate Change] – clear disambiguation of a novel core is paramount.

P6L15 – "These examples suggest that solar geoengineering would be more effective at changing surface melt than achieving the same reduction in temperature with a reduction in GHG forcing." – A fundamental assertion of this article is that SW reduction is more effective in modulating melt than a LW reduction, but there is a huge body of literature to suggest that melt is LW-dominated. To review Ohmura (2001; J. Applied Meteorology) – under cloudless-sky conditions, 90% of atmospheric emission is derived from the first 1 km of atmosphere – which is why air temperature index can perform remarkably well as a melt proxy. I am not sure that LW modification by GHG drawdown can be ignored entirely.

P6L20 – "The effect will be greatest for glaciers and ice sheets that are presently in negative mass balance and have the greatest amount of incoming solar radiation, that is glaciers at low latitudes such as in High Mountain Asia." My understanding is that the stratosphere is several km lower polar areas than at mid latitudes, so the majority of solar geoengineering proposals have advocating for injection aerosols into the polar regions. If this is indeed the case, I am not sure why low latitudes would benefit more from injected aerosols than high latitudes.

P7L20 – This discussion of ice dynamics should more clearly articulate the concept of committed mass loss. I suspect a quantitative assessment of solar geoengineering SMB buffering potential would find that committed loss from Antarctica is substantially larger. It may also be disingenuous to say that SW engineering could counter some of the ice dynamics trends now underway. The major mass loss contributors like Thwaites Glacier do not have ice shelves (i.e. Joughin et al., 2014; Science). The physical basis of committed mass loss purports that once it is triggered, it is only the density difference between ice and water, along with the gradient in bedrock slope, that determines when retreat will stop.

P2L29 – Scalable to 4W/m2. The potential magnitude of SW modification is never compared with characteristic magnitude for SMB components. Fausto (2016; GRL) presents a straightforward radiation balance associated with extreme melt events in

Greenland. The article would benefit from a simple thought experiment, whereby a plausible magnitude of SW RF suppression is applied to a summer melt season. The Fausto2016 values, for example have daily mean incoming SW around 100W/m2, with several instances of daily mean sensible heat flux exceed 50 W/m2. Without the authors saying what range of SW modification scenario they deem feasible, it is tough to gauge how that will ultimately effect melt.

Bioalbedo – If 4 W/m2 decreased incoming SW on a total incoming radiation of 150 W/m2 daily mean is being proposed, that is something like a 2.7% decrease in incident radiation. Emerging mechanisms are highlighting much larger changes in melt season albedo. For example, bioalbedo feedback (darkening of the glacier surface due to snow algae) can lower melt season albedo by 13% (or five times as much as the plausible SG mentioned in passing). This sort of contextualization of solar geoengineering is critical but absent from this paper. In jest, one could ask if cryospheric experts would better combat climate change by finding a "cure" for snow algae.

P3L10 – This discussion of the multifaceted effects of aerosol injection seems somewhat cursory/inferior to the tabulated pros and cons of Robock et al. (2009; GRL). I would also note a general absence of comparison with that study, which, for example, yields very different costs estimates of placing 1 Tg S in the stratosphere, and is generally much, much, more negative about the side-effects of geoengineering than presents here.

Section 5 – This section seems mislabeled as "sea-level rise engineering". One would expect that discussion to move towards how many mm sea-level equivalent may be associated with each geoengineered W/m2, instead this is rather a rehash candidate aerosols with the only tangential brush with sea level being discussion of seasonality of SMB modification.

Section 5 – This section is introduced as highlighting why it is "critical to introduce solar geoengineering into such analyses [of future sea level rise]" (P3 L30) – but does seems

to miss that mark. Pointing to an IPCC/EGU/EGU community statement on the value of solar geoengineering may serve to anchor the "critical" assertion, but my sense is that international reports generally do not advocate for the inclusion of solar geoengineering as "critical" (i.e. https://eos.org/agu-news/revised-agu-position-statement-addresses-climate-intervention) Perhaps an analogy is a small group of permafrost researchers saying the potential for an Arctic methane bomb is vastly more important than judged by the IPCC. OK, but why? Expand.

P6L33 – "As solar geoengineering would lower temperatures and reduce the intensity of the hydrological cycle it would reduce, perhaps even reverse, the negative contribution of Antarctic Surface Mass Balance to sea-level rise." May I highlight his sentence a microcosm of the paper? Unabashed praise for the promise of solar geoengineering with no apparent source for this tremendously speculative statement, and also glazes over/ignores a good deal of cryospheric research that highlights East Antarctic's SMB (the majority of the continent) is net positive, meaning it already draws down sea level today.

P11L9 – "Solar geoengineering could be deployed to not just reduce sea-level rise but to halt or even reverse it (Irvine et al. 2012)." This sentence is quite problematic. Irvine et al. (2012) only discuss the potential to stop sea-level rise, not reverse it as is being implied by this (self) citation. Keith and Irvine (2016) previous characterize the same study (Irvine et al., 2012) as demonstrating feasibility of solar geoengineering to limit sea-level rise "...by around a quarter". Highlighting these differences in self-characterization of previous studies makes me uneasy, as it seems the current manuscript could be used as a vehicle for expanding, without new foundation, the implications of earlier studies. Here, I caution the editors that it is difficult for me, or perhaps any reader, to comfortably separate conjecture from fact.

Summary: I might summarize this article as 60% non-cryosphere, which I am familiar with from previous studies, and 40% cryosphere, which I feel is not robust or up-to-date with the present literature. An idealized surface energy budget with and without solar

geoengineering modification seem like a minimum requirement to highlight precisely why solar geoengineering is "critical" for the cryospheric community to consider. I get the slight sense that the Brief Communication format here being used more like a popular opinion piece than a substantive review of the subject.
* * *

---

## Referee Comment (RC2) · Anonymous Referee #2 · 7 Feb 2018

This article reviews the links between solar radiation management (SRM) and the dynamic and surface mass balance (SMB) of ice sheets. However, there is no effort to understand or even reduce the uncertainty on the ice sheet component under SRM, except to provide an action plan to do this. The focus of climate modelers is on making future scenario based projections of sea level rise with new coupled ice sheet components. There is a long way to go before we can attempt to understand paleo-simulations much less SRM. Since the influence of SRM on ice sheet dynamics is unexplored, I would suggest the paper focus on SMB and should ideally include an analysis, however brief, of the GeoMIP model simulations. The article is bloated in comparison to what can be concluded from the small number of relevant simulations. In addition I find

some of the assertions at odds with the references omitted from this review, and these are commented on below.

P1:L28. You are referencing 'Expert Judgements' here, which do not really quantify projection uncertainty. The uncertainty should be expressed from model projections as described in AR5 (Ch 13). This is relevant since the next sentence refers to two such projections. P1:L29. Remove 'both of which were published in Nature'. This is a judgement statement implying quality of the referenced research (although this is not the use here, the commonality in source of the papers is irrelevant)! P1:L29-30. State the period at which these estimates of sea level equivalent apply. 2100? P1:L32-35. Evidence required. AR5 (Ch 12 & 13) provides this as does Bouttes (2013) below. Bouttes, N., J.M. Gregory, and J.A. Lowe, 2013: The Reversibility of Sea Level Rise. J. Climate, 26, 2502–2513, https://doi.org/10.1175/JCLI-D-12-00285.1 P2:L1. Carbon removal (e.g. Jones CD et al, 2016, Environ. Res. Lett. 11, 095012). P2:L29. RF and GHG not previously defined P4:L2-3. This is not self evident. Kravitz et al (2013) suggest that a polar warming might occur with over-cooling in the tropics, when compared against the reference state (Preindustrial). Kravitz, B., et al. (2013), Climate model response from the Geoengineering Model Intercomparison Project (GeoMIP), J. Geophys. Res. Atmos., 118, 8320–8332, doi:10.1002/jgrd.50646. P4:L9-15. Simple models do not show Greenland ice sheet decline for the strong climate mitigation scenario RCP2.6 either. P5:L3. Precipitation is decreased except for over the ice sheets (see fig 7 in Kavitz et al., 2013). P5:17-10. This is definitely not true. Nearly all modern Earth System Models now have a dynamic Greenland ice sheet and a few have mountain glaciers, and they are always, of course, driven by the ESM coupled fluxes (e.g. Lipsomb et al., 2013) . ISMIP6 is NOT using PPD for its offline models. Lipscomb, W.H., J.G. Fyke, M. Vizcaíno, W.J. Sacks, J. Wolfe, M. Vertenstein, A. Craig, E. Kluzek, and D.M. Lawrence, 2013: Implementation and Initial Evaluation of the Glimmer Community Ice Sheet Model in the Community Earth System Model. J. Climate, 26, 7352–7371, https://doi.org/10.1175/JCLI-D-12-00557.1 P6:L34. Actually, the hydrological cycle under SRM is increased over ice sheets (Kravitz et al., 2013). P7:L13.

Need to briefly state what "marine ice sheet instability" actually is. E.g. Grounding-line retreat leads to larger ice mass flux through the grounding-line generating further retreat. P7:L17 More precision, perhaps "They suggest that the atmospheric warming that led to the break-up of some Antarctic Peninsula ice shelves would, if the warming continued, destabilize the larger southern ice shelves in the future (Liu et al., 2015). The process is through the hydrostatic head of melt-water filled crevasses which results in "hydrofracture" and the rapid disintegration of the ice shelf." Though actually it is the Ice Cliff Instability (ICI) that is the killer in DeConto and Pollard but the ice shelves need to go first and in any case SRM will never stop ICI. Stick to the key point from this paper is that air temperatures are perhaps important for ice sheet collapse and these can easily be reversed. You are spending too much time on in DeConto and Pollard given the uncertainty they themselves express in the paper. You can be much briefer here. P8:L3-9. This whole discussion belongs back at the first paragraph of this section. Putting it here leads to a disjointed argument and repetition. Getting circumpolar water up on to the shelves depends on the Ekman pumping which is a function of the circumpolar winds. If the winds shift because of SRM or associated ozone depletion then the basal melt will be different. I have not seen any study of changes in the southern ocean winds under SRM. Intermediate waters are not going to cool significantly on the timescale SRM might be deployed. P9:L25. Bouttes et al., 2013 is relevant to this discussion. P10:L15-30. A few coupled global climate models are now including an interactive Antarctic and Greenland ice sheet components. Such models would enable a more complete understanding of the impact of SRM on ice sheets, than the doggy offline components.

---

## Author Comment (AC1) · 12 Jun 2018

A combined pdf with responses to both reviewers is attached. A - Author responses, R - Reviewer comments

A - We submitted this article to The Cryosphere as a brief communication after communicating to the editors that our article that was someway between a commentary and a technical review of the sea-level rise response to solar geoengineering. In such articles novelty is not the central goal. However, in responding to the reviewer comments we have added some novel analysis of the surface mass balance response to solar geoengineering. With the revisions recommended by the reviewers we believe

this article makes a useful contribution to the discussion on the sea-level rise response to solar geoengineering.

Reviewer 1

R - This article reviews the links between solar engineering and the surface mass balance of glaciers and ice sheets. Given the potential importance of the topic I am rather reluctant to report this article to be a rather awkward read. It pokes out in many directions, but not sufficiently far enough in any one to be truly novel. Perhaps this opinion reflects that I am well-read on the general topic, and personally feel that this qualitative discussion on cryospheric implications fall short of The Cryosphere community's consistent ability to deliver quantitative assessments on just about every other front. Personal opinion aside, this article objectively resurveys many of the same well-trodden roads of Irvine et al. [2017; Earth Future], Keith and Irvine [2016; Earth Future] and Irvine et al. [2012; Nature Climate Change] – clear disambiguation of a novel core is paramount.

A - We thank the reviewer for their suggestions and have made several major changes to address the concerns raised and to improve the manuscript: - We've added a quantitative analysis of the factors driving surface mass balance changes for the GeoMIP climate model ensemble. - We've restructured the main section of the paper. Sections 3 and 4 from the original paper are now sub-sections of a broader section which frames the issues we address more clearly and also briefly addresses thermosteric sea-level rise. - We've removed the "sea level rise engineering" section - We've rewritten the recommendations for research.

R - P6L15 – "These examples suggest that solar geoengineering would be more effective at changing surface melt than achieving the same reduction in temperature with a reduction in GHG forcing." – A fundamental assertion of this article is that SW reduction is more effective in modulating melt than a LW reduction, but there is a huge body of literature to suggest that melt is LW-dominated. To review Ohmura (2001; J.
Applied Meteorology) – under cloudless-sky conditions, 90% of atmospheric emission is derived from the first 1 km of atmosphere – which is why air temperature index can perform remarkably well as a melt proxy. I am not sure that LW modification by GHG drawdown can be ignored entirely.

A - Ohmura (2001) explains the surprisingly robust physical basis for the surface air temperature driven approaches to surface melt used in many models, highlighting the fact that surface air temperature is strongly correlated with lower atmospheric temperature and hence downwelling longwave (LW) radiation. Against a backdrop of elevated GHGs Solar geoengineering would cool the surface and lower atmosphere and so there will be a significant reduction in downwelling longwave (LW) radiation compared to a case with elevated GHGs and no solar geoengineering. The examples we highlight suggest that all-else-equal offsetting GHG forcing by a reduction in incoming sunlight would produce a greater reduction in melt than an equivalent reduction in CO2 forcing.

A - We note the insights of the Ohmura (2001) paper at the start of the surface mass balance section, changing the tone to be less critical of positive degree-day models of surface melt. The new quantitative analysis of the GeoMIP results, which we add at roughly the point the reviewer refers to, describes the changes in surface energy budget that bears out our intuition that solar geoengineering would be more effective at changing the surface energy budget than an equivalent reduction in GHG forcing.

R - P6L20 – "The effect will be greatest for glaciers and ice sheets that are presently in negative mass balance and have the greatest amount of incoming solar radiation, that is glaciers at low latitudes such as in High Mountain Asia." My understanding is that the stratosphere is several km lower polar areas than at mid latitudes, so the majority of solar geoengineering proposals have advocating for injection aerosols into the polar regions. If this is indeed the case, I am not sure why low latitudes would benefit more from injected aerosols than high latitudes.

A - The optical depth of the aerosol cloud will determine the fraction of light that it

scatters and hence the reduction in sunlight that reaches the surface below. Simply injecting aerosols into the equatorial stratosphere can produce a fairly evenly distributed global aerosol cloud with effects similar to a reduction in incoming sunlight (Niemeier et al. 2013, 10.1002/2013JD020445) though fine-tuning can produce a much more even cloud (Kravitz et al. 2018, 10.1002/2017JD026874). This means that all regions should experience a similar fractional change in incoming sunlight. The fact that the aerosol layer is at a lower altitude at high latitudes should not affect this.

A - We have edited this section to make clear that a fractional change in incoming sunlight at the ice surface will have a greater effect in sunnier places, i.e. lower latitude regions.

R - P2L29 – Scalable to 4W/m2. The potential magnitude of SW modification is never compared with characteristic magnitude for SMB components. Fausto (2016; GRL) presents a straightforward radiation balance associated with extreme melt events in Greenland. The article would benefit from a simple thought experiment, whereby a plausible magnitude of SW RF suppression is applied to a summer melt season. The Fausto2016 values, for example have daily mean incoming SW around 100W/m2, with several instances of daily mean sensible heat flux exceed 50 W/m2. Without the authors saying what range of SW modification scenario they deem feasible, it is tough to gauge how that will ultimately effect melt.

A - The significance of this quoted figure was perhaps not clear so we have added a note in the text that 4Wm-2 is roughly equal to the forcing from a doubling of CO2. We believe the new quantitative surface mass balance analysis of the GeoMIP ensemble addresses the reviewer's concern here.

R - Bioalbedo – If 4 W/m2 decreased incoming SW on a total incoming radiation of 150 W/m2 daily mean is being proposed, that is something like a 2.7% decrease in incident radiation. Emerging mechanisms are highlighting much larger changes in melt season albedo. For example, bioalbedo feedback (darkening of the glacier surface due to snow

algae) can lower melt season albedo by 13% (or five times as much as the plausible SG mentioned in passing). This sort of contextualization of solar geoengineering is critical but absent from this paper. In jest, one could ask if cryospheric experts would better combat climate change by finding a "cure" for snow algae.

A - We thank the reviewer for highlighting this omission, the darkening of snow by pollution and by snow algae is an important factor to consider. Snow surfaces with a lower albedo would exhibit a greater sensitivity to changes in incoming sunlight than brighter snow surfaces. This suggests that our quantitive results which focus on the responses over the entire ice-sheet may be underestimating the efficacy of solar geoengineering to reduce melt. We've added some text to explain how the response over darkened snow differs from that over fresh snow and clean ice.

R - P3L10 – This discussion of the multifaceted effects of aerosol injection seems somewhat cursory/inferior to the tabulated pros and cons of Robock et al. (2009; GRL). I would also note a general absence of comparison with that study, which, for example, yields very different costs estimates of placing 1 Tg S in the stratosphere, and is generally much, much, more negative about the side-effects of geoengineering than presents here.

A - We didn't believe that a full discussion of the pros and cons of solar geoengineering would be appropriate in a short-format article focused on the cryosphere response, and so included only a brief description of the major side-effects. In introducing the side-effects of stratospheric aerosol geoengineering, we now point the reader to a more up-to-date review of the full effects (Irvine et al. 2016, Wiley Interdisciplinary Reviews). We do not agree with the reviewer's assessment that we have underplayed the side-effects of stratospheric aerosol geoengineering, we believe the text adequately described most of these side-effects. For the shift from direct to diffuse light we've added a brief note on the implications of this shift for plant productivity and concentrating solar power. In terms of the costs of deployment, we refer to more recent estimates than that of Robock et al. (2009) and note that in personal communications Alan Robock agrees

with the newer estimates of the costs (personal communication between David Keith and Alan Robock).

R - Section 5 – This section seems mislabeled as "sea-level rise engineering". One would expect that discussion to move towards how many mm sea-level equivalent may be associated with each geoengineered W/m2, instead this is rather a rehash candidate aerosols with the only tangential brush with sea level being discussion of seasonality of SMB modification.

A - We have removed this section.

R - Section 5 – This section is introduced as highlighting why it is "critical to introduce solar geoengineering into such analyses [of future sea level rise]" (P3 L30) – but does seems to miss that mark. Pointing to an IPCC/EGU/EGU community statement on the value of solar geoengineering may serve to anchor the "critical" assertion, but my sense is that international reports generally do not advocate for the inclusion of solar geoengineering as "critical" (i.e. https://eos.org/agu-news/revised-agu-position-statement-addressesclimate-intervention) Perhaps an analogy is a small group of permafrost researchers saying the potential for an Arctic methane bomb is vastly more important than judged by the IPCC. OK, but why? Expand.

A - The description of the section in this paper that appeared on page 3 was from an earlier draft and did not reflect the structure of the piece we submitted. We no longer make this specific claim.

R - P6L33 – "As solar geoengineering would lower temperatures and reduce the intensity of the hydrological cycle it would reduce, perhaps even reverse, the negative contribution of Antarctic Surface Mass Balance to sea-level rise." May I highlight his sentence a microcosm of the paper? Unabashed praise for the promise of solar geoengineering with no apparent source for this tremendously speculative statement, and also glazes over/ignores a good deal of cryospheric research that highlights East Antarctic's SMB (the majority of the continent) is net positive, meaning it already draws down sea level

today.

A - Our text here was perhaps not as clear as it could have been. We were indeed referring to the net positive SMB of Antarctica today (which is a net negative contribution to SLR, as we noted) and suggesting that solar geoengineering could potentially offset or even reverse that. We make the same claim in the revised surface mass balance section:

A - "These results suggest that the negative contribution to sea-level rise of the positive surface mass balance response of Antarctica to global warming would decline roughly in line with temperatures if solar geoengineering were deployed though more work is needed to explore this issue."

R - P7L20 – This discussion of ice dynamics should more clearly articulate the concept of committed mass loss. I suspect a quantitative assessment of solar geoengineering SMB buffering potential would find that committed loss from Antarctica is substantially larger. It may also be disingenuous to say that SW engineering could counter some of the ice dynamics trends now underway. The major mass loss contributors like Thwaites Glacier do not have ice shelves (i.e. Joughin et al., 2014; Science). The physical basis of committed mass loss purports that once it is triggered, it is only the density difference between ice and water, along with the gradient in bedrock slope, that determines when retreat will stop.

A - We accept the reviewer's criticism on this point, we perhaps overstated the potential of solar geoengineering in this regard. We have completely rewritten the section (now section 3.3) and end with a more complete discussion of ice dynamics changes that stresses the committed mass loss.

R - P11L9 – "Solar geoengineering could be deployed to not just reduce sea-level rise but to halt or even reverse it (Irvine et al. 2012)." This sentence is quite problematic. Irvine et al. (2012) only discuss the potential to stop sea-level rise, not reverse it as is being implied by this (self) citation. Keith and Irvine (2016) previous characterize

the same study (Irvine et al., 2012) as demonstrating feasibility of solar geoengineering to limit sea-level rise "...by around a quarter". Highlighting these differences in self-characterization of previous studies makes me uneasy, as it seems the current manuscript could be used as a vehicle for expanding, without new foundation, the implications of earlier studies. Here, I caution the editors that it is difficult for me, or perhaps any reader, to comfortably separate conjecture from fact.

A - The reviewer is right that we make two different statements about the potential of solar geoengineering to change sea-level rise based on the results of a single paper. However, both are appropriate as they refer to different scenarios of solar geoengineering deployment. Irvine et al. (2012) analyzed solar geoengineering scenarios built off the RCP 8.5 emissions scenario with reductions in radiative forcing at 2100 ranging from 2.75 to 9.5 Wm 2, i.e. ranging from scenarios that reduce the warming by around a third (and sea-level rise by around a quarter) to scenarios that reduce temperatures below the pre-industrial mean (reversing recent sea-level rise in these simulations). However, in the revising the text we no longer make this specific claim.

R - Summary: I might summarize this article as 60% non-cryosphere, which I am familiar with from previous studies, and 40% cryosphere, which I feel is not robust or up-to-date with the present literature. An idealized surface energy budget with and without solar geoengineering modification seem like a minimum requirement to highlight precisely why solar geoengineering is "critical" for the cryospheric community to consider. I get the slight sense that the Brief Communication format here being used more like a popular opinion piece than a substantive review of the subject.

A - We have revised the paper substantially based on the reviewer's suggestions and hope that these changes address the concerns raised.

Please also note the supplement to this comment:
https://www.the-cryosphere-discuss.net/tc-2017-279/tc-2017-279-AC1-supplement.pdf

[Figure]

**Supplement:**

Response to reviews

We submitted this article to The Cryosphere as a brief communication after communicating to the editors that our article that was someway between a commentary and a technical review of the sea-level rise response to solar geoengineering. In such articles novelty is not the central goal. However, in responding to the reviewer comments we have added some novel analysis of the surface mass balance response to solar geoengineering. With the revisions recommended by the reviewers we believe this article makes a useful contribution to the discussion on the sea-level rise response to solar geoengineering.

Reviewer 1

This article reviews the links between solar engineering and the surface mass balance of glaciers and ice sheets. Given the potential importance of the topic I am rather reluctant to report this article to be a rather awkward read. It pokes out in many directions, but not sufficiently far enough in any one to be truly novel. Perhaps this opinion reflects that I am well-read on the general topic, and personally feel that this qualitative discussion on cryospheric implications fall short of The Cryosphere community's consistent ability to deliver quantitative assessments on just about every other front. Personal opinion aside, this article objectively resurveys many of the same well-trodden roads of Irvine et al. [2017; Earth Future], Keith and Irvine [2016; Earth Future] and Irvine et al. [2012; Nature Climate Change] – clear disambiguation of a novel core is paramount.

We thank the reviewer for their suggestions and have made several major changes to address the concerns raised and to improve the manuscript:
- We've added a quantitative analysis of the factors driving surface mass balance changes for the GeoMIP climate model ensemble.
- We've restructured the main section of the paper. Sections 3 and 4 from the original paper are now sub-sections of a broader section which frames the issues we address more clearly and also briefly addresses thermosteric sea-level rise.
- We've removed the "sea level rise engineering" section
- We've rewritten the recommendations for research.

P6L15 – "These examples suggest that solar geoengineering would be more effective at changing surface melt than achieving the same reduction in temperature with a reduction in GHG forcing." – A fundamental assertion of this article is that SW reduction is more effective in modulating melt than a LW reduction, but there is a huge body of literature to suggest that melt is LW-dominated. To review Ohmura (2001; J. Applied Meteorology) – under cloudless-sky conditions, 90% of atmospheric emission is derived from the first 1 km of atmosphere – which is why air temperature index can perform remarkably well as a melt proxy. I am not sure that LW modification by GHG drawdown can be ignored entirely.

Ohmura (2001) explains the surprisingly robust physical basis for the surface air temperature driven approaches to surface melt used in many models, highlighting the fact that surface air temperature is strongly correlated with lower atmospheric temperature and hence downwelling longwave (LW) radiation. Against a backdrop of elevated GHGs Solar geoengineering would cool the surface and lower atmosphere and so there will be a significant reduction in downwelling longwave (LW) radiation compared to a case with elevated GHGs and no solar geoengineering. The examples we highlight

suggest that all-else-equal offsetting GHG forcing by a reduction in incoming sunlight would produce a greater reduction in melt than an equivalent reduction in $CO_2$ forcing.

We note the insights of the Ohmura (2001) paper at the start of the surface mass balance section, changing the tone to be less critical of positive degree-day models of surface melt. The new quantitative analysis of the GeoMIP results, which we add at roughly the point the reviewer refers to, describes the changes in surface energy budget that bears out our intuition that solar geoengineering would be more effective at changing the surface energy budget than an equivalent reduction in GHG forcing.

P6L20 – "The effect will be greatest for glaciers and ice sheets that are presently in negative mass balance and have the greatest amount of incoming solar radiation, that is glaciers at low latitudes such as in High Mountain Asia." My understanding is that the stratosphere is several km lower polar areas than at mid latitudes, so the majority of solar geoengineering proposals have advocating for injection aerosols into the polar regions. If this is indeed the case, I am not sure why low latitudes would benefit more from injected aerosols than high latitudes.

The optical depth of the aerosol cloud will determine the fraction of light that it scatters and hence the reduction in sunlight that reaches the surface below. Simply injecting aerosols into the equatorial stratosphere can produce a fairly evenly distributed global aerosol cloud with effects similar to a reduction in incoming sunlight (Niemeier et al. 2013, 10.1002/2013JD020445) though fine-tuning can produce a much more even cloud (Kravitz et al. 2018, 10.1002/2017JD026874). This means that all regions should experience a similar fractional change in incoming sunlight. The fact that the aerosol layer is at a lower altitude at high latitudes should not affect this.

We have edited this section to make clear that a fractional change in incoming sunlight at the ice surface will have a greater effect in sunnier places, i.e. lower latitude regions.

P2L29 – Scalable to 4W/m2. The potential magnitude of SW modification is never compared with characteristic magnitude for SMB components. Fausto (2016; GRL) presents a straightforward radiation balance associated with extreme melt events in Greenland. The article would benefit from a simple thought experiment, whereby a plausible magnitude of SW RF suppression is applied to a summer melt season. The Fausto2016 values, for example have daily mean incoming SW around 100W/m2, with several instances of daily mean sensible heat flux exceed 50 W/m2. Without the authors saying what range of SW modification scenario they deem feasible, it is tough to gauge how that will ultimately effect melt.

The significance of this quoted figure was perhaps not clear so we have added a note in the text that 4Wm-2 is roughly equal to the forcing from a doubling of $CO_2$. We believe the new quantitative surface mass balance analysis of the GeoMIP ensemble addresses the reviewer's concern here.

Bioalbedo – If 4 W/m2 decreased incoming SW on a total incoming radiation of 150 W/m2 daily mean is being proposed, that is something like a 2.7% decrease in incident radiation. Emerging mechanisms are highlighting much larger changes in melt season albedo. For example, bioalbedo feedback (darkening of the glacier surface due to snow algae) can lower melt season albedo by 13% (or five times as much as the plausible SG mentioned in passing). This sort of contextualization of solar geoengineering is critical but absent from this paper. In jest, one could ask if cryospheric experts would better combat climate change by finding a "cure" for snow algae.

We thank the reviewer for highlighting this omission, the darkening of snow by pollution and by snow algae is an important factor to consider. Snow surfaces with a lower albedo would exhibit a greater sensitivity to changes in incoming sunlight than brighter snow surfaces. This suggests that our quantitive results which focus on the responses over the entire ice-sheet may be underestimating the efficacy of solar geoengineering to reduce melt. We've added some text to explain how the response over darkened snow differs from that over fresh snow and clean ice.

P3L10 – This discussion of the multifaceted effects of aerosol injection seems somewhat cursory/inferior to the tabulated pros and cons of Robock et al. (2009; GRL). I would also note a general absence of comparison with that study, which, for example, yields very different costs estimates of placing 1 Tg S in the stratosphere, and is generally much, much, more negative about the side-effects of geoengineering than presents here.

We didn't believe that a full discussion of the pros and cons of solar geoengineering would be appropriate in a short-format article focused on the cryosphere response, and so included only a brief description of the major side-effects. In introducing the side-effects of stratospheric aerosol geoengineering, we now point the reader to a more up-to-date review of the full effects (Irvine et al. 2016, Wiley Interdisciplinary Reviews). We do not agree with the reviewer's assessment that we have underplayed the side-effects of stratospheric aerosol geoengineering, we believe the text adequately described most of these side-effects. For the shift from direct to diffuse light we've added a brief note on the implications of this shift for plant productivity and concentrating solar power. In terms of the costs of deployment, we refer to more recent estimates than that of Robock et al. (2009) and note that in personal communications Alan Robock agrees with the newer estimates of the costs (personal communication between David Keith and Alan Robock).

Section 5 – This section seems mislabeled as "sea-level rise engineering". One would expect that discussion to move towards how many mm sea-level equivalent may be associated with each geoengineered W/m2, instead this is rather a rehash candidate aerosols with the only tangential brush with sea level being discussion of seasonality of SMB modification.

We have removed this section.

Section 5 – This section is introduced as highlighting why it is "critical to introduce solar geoengineering into such analyses [of future sea level rise]" (P3 L30) – but does seems to miss that mark. Pointing to an IPCC/EGU/EGU community statement on the value of solar geoengineering may serve to anchor the "critical" assertion, but my sense is that international reports generally do not advocate for the inclusion of solar geoengineering as "critical" (i.e. https://eos.org/agu-news/revised-agu-position-statement-addressesclimate-intervention) Perhaps an analogy is a small group of permafrost researchers saying the potential for an Arctic methane bomb is vastly more important than judged by the IPCC. OK, but why? Expand.

The description of the section in this paper that appeared on page 3 was from an earlier draft and did not reflect the structure of the piece we submitted. We no longer make this specific claim.

P6L33 – "As solar geoengineering would lower temperatures and reduce the intensity of the hydrological cycle it would reduce, perhaps even reverse, the negative contribution of Antarctic Surface Mass Balance to sea-level rise." May I highlight his sentence a microcosm of the paper? Unabashed praise for the promise of solar geoengineering with no apparent source for this tremendously

speculative statement, and also glazes over/ignores a good deal of cryospheric research that highlights East Antarctic's SMB (the majority of the continent) is net positive, meaning it already draws down sea level today.

Our text here was perhaps not as clear as it could have been. We were indeed referring to the net positive SMB of Antarctica today (which is a net negative contribution to SLR, as we noted) and suggesting that solar geoengineering could potentially offset or even reverse that. We make the same claim in the revised surface mass balance section:

"These results suggest that the negative contribution to sea-level rise of the positive surface mass balance response of Antarctica to global warming would decline roughly in line with temperatures if solar geoengineering were deployed though more work is needed to explore this issue."

P7L20 – This discussion of ice dynamics should more clearly articulate the concept of committed mass loss. I suspect a quantitative assessment of solar geoengineering SMB buffering potential would find that committed loss from Antarctica is substantially larger. It may also be disingenuous to say that SW engineering could counter some of the ice dynamics trends now underway. The major mass loss contributors like Thwaites Glacier do not have ice shelves (i.e. Joughin et al., 2014; Science). The physical basis of committed mass loss purports that once it is triggered, it is only the density difference between ice and water, along with the gradient in bedrock slope, that determines when retreat will stop.

We accept the reviewer's criticism on this point, we perhaps overstated the potential of solar geoengineering in this regard. We have completely rewritten the section (now section 3.3) and end with a more complete discussion of ice dynamics changes that stresses the committed mass loss.

P11L9 – "Solar geoengineering could be deployed to not just reduce sea-level rise but to halt or even reverse it (Irvine et al. 2012)." This sentence is quite problematic. Irvine et al. (2012) only discuss the potential to stop sea-level rise, not reverse it as is being implied by this (self) citation. Keith and Irvine (2016) previous characterize the same study (Irvine et al., 2012) as demonstrating feasibility of solar geoengineering to limit sea-level rise "…by around a quarter". Highlighting these differences in self-characterization of previous studies makes me uneasy, as it seems the current manuscript could be used as a vehicle for expanding, without new foundation, the implications of earlier studies. Here, I caution the editors that it is difficult for me, or perhaps any reader, to comfortably separate conjecture from fact.

The reviewer is right that we make two different statements about the potential of solar geoengineering to change sea-level rise based on the results of a single paper. However, both are appropriate as they refer to different scenarios of solar geoengineering deployment. Irvine et al. (2012) analyzed solar geoengineering scenarios built off the RCP 8.5 emissions scenario with reductions in radiative forcing at 2100 ranging from 2.75 to 9.5 Wm-2, i.e. ranging from scenarios that reduce the warming by around a third (and sea-level rise by around a quarter) to scenarios that reduce temperatures below the pre-industrial mean (reversing recent sea-level rise in these simulations). However, in the revising the text we no longer make this specific claim.

Summary: I might summarize this article as 60% non-cryosphere, which I am familiar with from previous studies, and 40% cryosphere, which I feel is not robust or up-to-date with the present literature. An idealized surface energy budget with and without solar geoengineering modification seem like a

minimum requirement to highlight precisely why solar geoengineering is "critical" for the cryospheric community to consider. I get the slight sense that the Brief Communication format here being used more like a popular opinion piece than a substantive review of the subject.

We have revised the paper substantially based on the reviewer's suggestions and hope that these changes address the concerns raised.

Reviewer 2

This article reviews the links between solar radiation management (SRM) and the dynamic and surface mass balance (SMB) of ice sheets. However, there is no effort to understand or even reduce the uncertainty on the ice sheet component under SRM, except to provide an action plan to do this. The focus of climate modelers is on making future scenario based projections of sea level rise with new coupled ice sheet components. There is a long way to go before we can attempt to understand paleo-simulations much less SRM. Since the influence of SRM on ice sheet dynamics is unexplored, I would suggest the paper focus on SMB and should ideally include an analysis, however brief, of the GeoMIP model simulations. The article is bloated in comparison to what can be concluded from the small number of relevant simulations. In addition I find some of the assertions at odds with the references omitted from this review, and these are commented on below.

We thank the reviewer for their suggestions and have made several major changes to address the concerns raised and to improve the manuscript:
- We've added a quantitative analysis of the factors driving surface mass balance changes for the GeoMIP climate model ensemble.
- We've restructured the main section of the paper. Sections 3 and 4 from the original paper are now sub-sections of a broader section which frames the issues we address more clearly and also briefly addresses thermosteric sea-level rise.
- We've removed the "sea level rise engineering" section
- We've rewritten the recommendations for research.

P1:L28. You are referencing 'Expert Judgements' here, which do not really quantify projection uncertainty. The uncertainty should be expressed from model projections as described in AR5 (Ch 13). This is relevant since the next sentence refers to two such projections.

Bayesian statistics is widely applied in the Earth sciences and in sea-level rise projections and provides a framework in which expert judgements can be used alongside other inputs to estimate uncertainty in projections. We believe that it is appropriate to refer to studies which draw on expert judgements of projection uncertainty in this case due to the fact all models miss certain processes which are known to be critical to the future contribution of ice-sheets to sea-level rise. To rely on the spread in model projections alone would be to severely under-estimate uncertainty in ice-sheet contributions to sea-level rise. As our point in this paragraph is to highlight the large uncertainty in sea-level rise contributions from Antarctica we believe it is appropriate to cite studies that illustrate this point using a range of approaches including expert judgement.

P1:L29. Remove 'both of which were published in Nature'. This is a judgement statement implying quality of the referenced research (although this is not the use here, the commonality in source of the papers is irrelevant)!

We've rephrased this as follows:

> "For example, two recent high-profile publications made conflicting estimates of Antarctica's contribution to sea-level rise by 2100 with a best-guess of 10cm (Ritz et al., 2015), and of around 1m (DeConto and Pollard, 2016)."

P1:L29-30. State the period at which these estimates of sea level equivalent apply. 2100?

See last response

P1:L32-35. Evidence required. AR5 (Ch 12 & 13) provides this as does Bouttes (2013) below. Bouttes, N., J.M. Gregory, and J.A. Lowe, 2013: The Reversibility of Sea Level Rise. J. Climate, 26, 2502–2513, https://doi.org/10.1175/JCLI-D-12-00285.1 P2:L1. Carbon removal (e.g. Jones CD et al, 2016, Environ. Res. Lett. 11, 095012).

We thank the reviewer for this useful suggestion which we cite elsewhere, though in this case we have cited Clark et al. (2016) which points out the millennial sea-level rise implications of fossil-fuel emissions (without CDR or solar geoengineering).

P2:L29. RF and GHG not previously defined

We have removed RF as this was the only usage and defined GHG here.

P4:L2-3. This is not self evident. Kravitz et al (2013) suggest that a polar warming might occur with over-cooling in the tropics, when compared against the reference state (Preindustrial). Kravitz, B., et al. (2013), Climate model response from the Geoengineering Model Intercomparison Project (GeoMIP), J. Geophys. Res. Atmos., 118, 8320–8332, doi:10.1002/jgrd.50646.

We have made it clearer in the text that we are referring to the effects of solar geoengineering alone, which cools everywhere, not the combined effect of elevated $CO_2$ and solar geoengineering. The relevant comparison in that Kravitz study is the abrupt4x$CO_2$ experiment, not the pre-industrial.

> "As solar geoengineering would reduce temperatures across the world, **offsetting some of the warming from elevated GHG concentrations,** it is clear that to first order it would reduce both the thermal expansion of the oceans and the melting of land ice."

P4:L9-15. Simple models do not show Greenland ice sheet decline for the strong climate mitigation scenario RCP2.6 either.

We've clarified that we are referring to high-$CO_2$ scenarios here.

P5:L3. Precipitation is decreased except for over the ice sheets (see fig 7 in Kavitz et al., 2013).

We pick this issue up in the revised section on surface mass balance.

P5:17-10. This is definitely not true. Nearly all modern Earth System Models now have a dynamic Greenland ice sheet and a few have mountain glaciers, and they are always, of course, driven by the ESM coupled fluxes (e.g. Lipsomb et al., 2013) . ISMIP6 is NOT using PPD for its offline models. Lipscomb,

W.H., J.G. Fyke, M. Vizcaíno, W.J. Sacks, J. Wolfe, M. Vertenstein, A. Craig, E. Kluzek, and D.M. Lawrence, 2013: Implementation and Initial Evaluation of the Glimmer Community Ice Sheet Model in the Community Earth System Model. J. Climate, 26, 7352–7371, https://doi.org/10.1175/JCLI-D-12-00557.1

We thank the reviewer for the correction, our view on this was shaped by our analysis of CMIP5-era models which, as IPCC AR5 WG1 Ch13 p1169, makes clear did not include coupled ice sheets: "Goelzer et al. (2013) and Gillet-Chaulet et al. (2012) suggested that SMB and ice dynamics cannot be assessed separately because of the strong interaction between ice loss and climate due to, for instance, calving and SMB. The current assessment has by necessity separated these effects because the type of coupled ice sheet-climate models needed to make a full assessment do not yet exist."

We have reworded the paragraph as follows:

"Many ice-sheet and glacier models use a simple parameterization of surface mass balance, using a positive degree-day factor to estimate the amount of melt per degree above freezing at the glacier surface (Ohmura, 2001). Degree day factors are determined empirically and vary due to surface albedo, meaning that a weathered ice surface such as the Greenland ice margin are rather dark and have high degree-day factors, while pristine snow cover has a low factor. This degree-day approach has been used in all studies of solar geoengineering's effect on surface mass balance to date, but it has some important limitations."

P6:L34. Actually, the hydrological cycle under SRM is increased over ice sheets (Kravitz et al., 2013).

We have rewritten this section and include results that support the reviewer's assessment.

P7:L13. Need to briefly state what "marine ice sheet instability" actually is. E.g. Grounding-line retreat leads to larger ice mass flux through the grounding-line generating further retreat.

This section has been completely rewritten (now section 3.3) and we include a brief description of marine ice sheet instability.

P7:L17 More precision, perhaps "They suggest that the atmospheric warming that led to the break-up of some Antarctic Peninsula ice shelves would, if the warming continued, destabilize the larger southern ice shelves in the future (Liu et al., 2015). The process is through the hydrostatic head of melt-water filled crevasses which results in "hydrofracture" and the rapid disintegration of the ice shelf." Though actually it is the Ice Cliff Instability (ICI) that is the killer in DeConto and Pollard but the ice shelves need to go first and in any case SRM will never stop ICI. Stick to the key point from this paper is that air temperatures are perhaps important for ice sheet collapse and these can easily be reversed. You are spending too much time on in DeConto and Pollard given the uncertainty they themselves express in the paper. You can be much briefer here.

We have revised this section considerably, reducing the amount of material on the DeConto and Pollard paper and focusing on the potential significance of surface air temperature on ice-shelf stability. We have adopted the phrasing suggested by the reviewer for those sentences.

P8:L3-9. This whole discussion belongs back at the first paragraph of this section. Putting it here leads to a disjointed argument and repetition. Getting circumpolar water up on to the shelves depends on the Ekman pumping which is a function of the circumpolar winds. If the winds shift because of SRM or

associated ozone depletion then the basal melt will be different. I have not seen any study of changes in the southern ocean winds under SRM. Intermediate waters are not going to cool significantly on the timescale SRM might be deployed.

We thank the reviewer for this useful suggestion. We have restructured and rewritten this section, brining this point up nearer to the beginning of this section.

P9:L25. Bouttes et al., 2013 is relevant to this discussion.

We thank the reviewer for this suggestion and cite Bouttes et al. (2013) on the reversibility of thermosteric sea-level rise in the new sub-section (3.1) devoted to this issue.

P10:L15-30. A few coupled global climate models are now including an interactive Antarctic and Greenland ice sheet components. Such models would enable a more complete understanding of the impact of SRM on ice sheets, than the doggy offline components.

We thank the reviewer for this suggestion and in the fully revised research recommendations sections, this is our first recommendation.

---

## Author Response (AR2)

**Editor:**

Dear Authors,

Thanks for the revised version of your paper which has been significantly improved thanks to both reviewers. As both of them are happy with your revised version, it is a pleasure for me to accept your paper for final publication in TC.

However, could you, before uploading final files, make the minor changes requested by reviewers ? There is also the question of the manuscript type listed here: https://www.the-cryosphere.net/about/manuscript_types.html

I think that your paper fits indeed better to "Brief Communication: (b) report/discuss on significant matters of policy and perspective related to the science of the journal, including "personal" commentary" but your paper is perhaps too long for this. Therefore I suggest you to add Brief Communication in your title according to this manuscript type and we will see with the Copernicus editorial team if they are Ok to keep your paper in this format although you have more than 20 references.

Best regards,

Xavier Fettweis

Dear Xavier Fettweis,

Thanks for coordinating this. We've made some of the suggested changes and argued why we haven't made the others below.

Thanks,

Pete Irvine

**Referee #1:**

Clearly, much work has been invested in this manuscript since its initial submission. In this regard, the GeoMIP simulations presented in section 3.2 are refreshingly novel and quantitative.

At this point, my primary reservation is that the authors seem to have rebutted the majority of R1 comments outside the main manuscript. It is not rebuttals themselves, but rather that this relevant content is not available to readers whom might have similar questions. For example, while I am pleased to see Ohmura2001 appear in the manuscript, what about contextualizing 4 W/m2 against characteristic surface energy budget terms? Or what about explicitly saying these per Tg SO4 cost estimates are different than those of Robuck et al. 2009? Why not mention that you are aware of differences in aerosol injection (and cloud) heights and properties between mid latitude and Arctic? These insights provide little service to the general readership when tucked away in the rebuttal letter.

Robock et al. 2009's cost estimates do not differ from those presented here, with an estimated annual deployment cost of $0.225 Billion to $4.175 Billion per year per Mt, and so we have added this reference to this list.

We provide only a very brief description of stratospheric aerosol geoengineering due to limited space and a focus on its potential cryosphere effects. As we argued, the altitude of the aerosol cloud does not matter for the purposes here and so was excluded from our brief survey. Motivated readers can refer to the cited studies to find these and many other details about stratospheric aerosol geoengineering. We have rephrased a few sentences to provide a little more detail though:

"Releasing a few Terragrams of material per year into the lower Tropical stratosphere (~20km) would produce an aerosol layer with global coverage. Multiple, independent feasibility assessments of the proposal conclude that this could be achieved at a cost of order one billion US dollars per Terragram using high-altitude jets (McClellan et al., 2012; Moriyama et al., 2016; Robock et al. 2009)."

We do not believe that the right comparison is between the global radiative forcing from solar geoengineering and the characteristic local surface energy budget terms which is why we have not included this. That the total global radiative forcing for both the GHG warming and solar geoengineering scenarios is perhaps only equal to 4% of the local incoming shortwave at some location is not the most important point, rather we believe the differences between the effects of these forcings is what matters most. Our quantitative evaluation of the differences between the effects of these two forcings in the surface mass balance section.

My secondary reservation remains fit with journal, or at least article format. The authors have responded to this saying the article "was someway between a commentary and a technical review." In terms of format, I do not see Brief Communications as a venue for review of any type. By avoiding a full length TC article, the authors skirt a greater onus on thoroughness and detail. But clearly the editorship has invited a revision within this format. In terms of journal fit, I think the stated sentiment that "novelty is not the central goal" really runs counter to The Cryosphere ambition. The inherent challenge of inter-disciplinary publications is to be simultaneously relevant and up-to-date with multiple communities. In this I can see that the authors have chosen no small task.

We believe the editor is satisfied with the paper as is.

**Referee #2:**

11-13: The sentence should be reversed. Mention cyrosphere and then melt otherwise melt is not in immediate context.

Done

: ...ability to reverse ...

Both comments are addressed in new phrasing: "The efficacy of solar geoengineering at reducing changes to the cryosphere is uncertain; solar geoengineering could reduce temperatures and so slow melt, but its ability to reverse ice sheet collapse once initiated may be limited."

: models

Rephrased to: "Studies of natural analogues and model simulations support this conclusion."

358-360 : Misleading. The BISICLES experiment was only reversed (from a stable state) because the entire water column was instantaneously cooled. As written a reader might assume it was just surface cooling.

We have addressed this and rephrased this as follows:

"However, initial results from the BISICLES model evaluating the response of an idealized vulnerable marine glacier to imposed warming found that returning the entire water column to cooler conditions reversed the retreat that had begun during the warming (Asay-Davis et al., 2016). It seems reasonable to expect that solar geoengineering, like emissions cuts, may help to prevent other marine glaciers from becoming unstable by limiting surface melt that could lead to ice-shelf collapse but would have a limited ability to reverse sub-surface warming on decadal timescales."

**Brief Communication:** **Understanding solar geoengineering's potential to limit sea level rise requires**
**attention from cryosphere experts**

Peter J. Irvine[1], David W. Keith[1], John Moore[2,3]

1 - Harvard John A. Paulson School of Engineering and Applied Sciences, Cambridge, Massachusetts, USA

2 - Joint Center for Global Change Studies, College of Global Change and Earth System Science, Beijing Normal
University, Beijing 100875, China

3 - Arctic Centre, University of Lapland, Rovaniemi 96101, Finland

**Abstract**

Stratospheric aerosol geoengineering, a form of solar geoengineering, is a proposal to add a reflective layer of
aerosol to the stratosphere to reduce net radiative forcing and so to reduce the risks of climate change. The efficacy
of solar geoengineering at reducing changes to the cryosphere is uncertain; solar geoengineering could reduce
temperatures and so slow melt, but
its ability to reverse ice sheet collapse once initiated may be limited. Here we review the literature on
solar geoengineering and the cryosphere and identify the key uncertainties that research could address. Solar
geoengineering may be more effective at reducing surface melt than a reduction in greenhouse forcing that produces
the same global-average temperature response. Studies of natural analogues and model simulations support this
conclusion. However, changes below the surfaces of the ocean and ice-sheets may strongly limit the potential of
solar geoengineering to reduce the retreat of marine glaciers. High-quality process model studies may illuminate
these questions. Solar geoengineering is a contentious emerging issue in climate policy and it is critical that the
potential, limits and risks of these proposals are made clear for policy makers.

**1. Future Sea-level rise and the potential of solar geoengineering**

How far sea-levels would rise under some scenario of future climate change depends mainly on global temperature
rise, and uncertainties in projections rise rapidly as warming increases more than 2°C above pre-industrial (Jevrejeva
et al., 2016; Kopp et al., 2014). Most of this uncertainty is due to a lack of agreement on how the large ice sheets
will respond (Bamber and Aspinall, 2013; Oppenheimer et al., 2016). For example, two recent high-profile
publications made conflicting estimates of Antarctica's contribution to sea-level rise by 2100 with a best-guess of
10cm (Ritz et al., 2015), and of around 1m (DeConto and Pollard, 2016).

A rapid transition towards a carbon-free economy will reduce additional temperature increases but the temperature
response to cumulative emissions—and thus the impact on sea level—will remain for millennia without measures
beyond emissions cuts (Clark et al., 2016). Two broad categories of measures might reduce long-term commitments
to global sea level rise: solar geoengineering and atmospheric carbon removal. Solar geoengineering which
describes a set of proposals to increase Earth's albedo, is not a substitute for emissions cuts. But it could offer an
independent means of temporarily reducing radiative forcing and thus the impacts of climate change, and so be a
complement to emissions cuts. The two responses may be synergistic: carbon removal can reduce the long-term
driver of climate change, while solar geoengineering might temporarily reduce the net radiative forcing. Our focus is
on assessing solar geoengineering impact on sea level rise because existing research is quite limited and because its
effects (per unit temperature change) may not be the same as those achieved by reducing temperature by de-
carbonizing.

The human, environmental and financial costs of sea level rise are substantial.  The rapidly rising concentration of
population and infrastructure in coastal cities mean that costs of flooding without adaptation measures are projected
to be $50 trillion per year by 2100, while coastal protection would cost $15-70 billion per year (Hinkel et al., 2014).
One important consideration is that sea level rise is not globally uniform, due to a combination of local factors:
glacial isostatic adjustment and ground water extraction resulting in local vertical land movement; the self-
gravitational influence of mass loss from the large ice sheets; and changes in ocean dynamics and rates of volume
expansion of warming sea water. Taking all these together, Jevrejeva et al. (2016) find that the 80-90% of global
coastlines will experience sea level rises about twice as large as the global ocean average.

Whilst some, including one of us (Keith), have been working on solar geoengineering for decades, more than ten
times as many articles have been published on the topic since 2007 than before. Whilst many proposals for solar
geoengineering have been made, work now focuses on a few of the more likely candidates. Marine Cloud
Brightening, a proposal to increase the albedo of marine strato-cumulus by releasing sea-salt aerosols from ships
(Latham, 1990); Cirrus Cloud Thinning, a proposal to suppress cirrus cloud persistence, and hence reduce their
warming effect, by releasing ice nuclei to encourage the formation of larger, shorter-lived ice crystals (Mitchell and
Finnegan, 2009); and Stratospheric Aerosol Geoengineering, a proposal to release aerosol particles into the
stratosphere to create a persistent reflective aerosol layer scattering a small fraction of incoming light back to space
(Budyko, 1977). Of these proposals stratospheric aerosol geoengineering is the most likely to be technically
achievable. ~Releasing a few Terragrams of material per year into the lower Tropical stratosphere (~20km) would

[revised manuscript text omitted]